# MLBF-PRS: A MACHINE LEARNING MODEL DEVELOPMENT AND BENCHMARKING FRAMEWORK FOR POLYGENIC RISK SCORES

## ABSTRACT

In contrast to other genomic tasks, the development of machine learning-based individual-level, genome-wide predictive models, typically termed polygenic risk scores (PRS), have shown little improvement from the use of complex machine learning (ML) methods. This disparity can be attributed to challenges in accessibility, comparability across studies, and a lack of development and evaluation guidelines that enable reproducibility. Sequence-based genomic tasks benefit from benchmarks, which have proven to be fruitful in the advancement of machine learning model development across domains.

To overcome the challenges present in the development of ML-based PRS models, we introduce MLBF-PRS, a novel framework as a catalyst to promote and accelerate the development of ML-based solutions. The framework provides flexible Nextflow DSL2 pipelines that enable parallel comparison of ML models (SVMs, random forests, neural networks) against established statistical PRS methods, comprehensive quality control and data preparation modules following PRS-specific best practices, and automated tracking of model parameters, trained weights, and configurations to ensure full reproducibility.

We describe the usage of MLBF-PRS to showcase how this framework provides accessibility, where, in most cases, the setup and evaluation of PRS models can be time-consuming and require navigation of multiple software tools. The standardised and reproducible dataset-specific benchmarking through MLBF-PRS offers a practical alternative to traditional open benchmarks. We make our framework openly available and continue expanding its capabilities.

## 1 INTRODUCTION

Polygenic risk scores (PRS) estimate an individual's population level risk of disease by accumulating the effect of multiple variations, single-nucleotide polymorphisms (SNPs), across the genome. In most cases these SNPs are substitutions of a single nucleotide (A,C,T,G) on the genome. The classical PRS of an individual $j$ in a population is defined as the sum over the number of affected alleles of the SNPs of interest multiplied by their respective effect sizes:

$$PRS_j = \sum_{i=1}^{N} \beta_i d_i, \qquad (1)$$

where $\beta_i \in \mathbb{R}$, $d_i \in \{0, 1, 2\}$ and $N \in \mathbb{N}_+$ the number of SNPs. The effect sizes in the classical model are derived from genome-wide association studies (GWAS). To this day, a multitude of extensions to the classical model have been developed (Ndong Sima et al., 2024; Benoumhani et al., 2025). Contrary to other genomic data tasks such as variant classification, functional annotation or DNA generation (Cheng et al., 2023; Avsec et al., 2025; Brixi et al., 2025), state-of-the-art models in PRS estimation do not rely on machine learning algorithms. These aforementioned genomic data machine learning models use genome sequence data, which differs quite significantly from the genotype array data used for PRS development, for which the data consists of single measurements of variations at single loci across the whole genome. Although there are ongoing efforts to develop

machine learning-based PRS, the adaptation of machine learning in PRS development significantly lags behind the aforementioned domains.

## 1.1 DATA ACCESSIBILITY

One obstacle to the development of machine learning-based PRS models exists due to data accessibility restrictions, which is a common issue with medical data that directly influences reproducibility (McDermott et al., 2021; McDermott, 2025). For genomic data problems involving genomic sequences, this issue can partially be circumvented through the use of reference genomes and open datasets. However, the evaluation and development of machine learning-based PRS methods in most cases requires access to individual level data from cohorts (Wang et al., 2023), such as the popular resource UK Biobank (Bycroft et al., 2018). Access to such informative and rich resources is subject to individual approval and data is not publicly available due to ethical guidelines, because individual level data contains sensitive information about study participants. This can slow down model development and stands in stark contrast to open benchmarking datasets or competitions commonly found in machine learning model development. At the same time, this data is necessary to improve the clinical applicability of a PRS. However, development of methodological concepts can proceed on synthetic datasets (Su et al., 2011; Tang & Liu, 2019; Wharrie et al., 2023), where genotypes and observable traits, phenotypes, are simulated based on openly available reference genomes such as the 1000 genomes (Auton et al., 2015). Nevertheless, synthetic data cannot fully substitute real data. Since data generation is controlled, this prior knowledge about the data can introduce bias in model development.

## 1.2 REQUIREMENTS FOR MACHINE LEARNING MODEL DEVELOPMENT

A recent survey of advancements in the development of deep learning-based PRS methods highlighted difficulties in comparability and reproducibility across phenotypes and datasets (Schuran et al., 2025). Dedicated deep learning architectures for PRS prediction were generally not evaluated against another in model development and the majority of publications reported favourable results towards their own models. A meta analysis of predictive performances is made difficult due to the evaluation of models on differing datasets in term of population structure, data size, features, and different outcomes of interest. In the analyses concerning the development and evaluation of deep learning-based models we can observe conflicting results concerning predictive performance improvements with respect to statistical models. Due to a lack of a common benchmark for machine learning-based models no definitive conclusion on advantages through machine learning models can be made. We attribute the absence of such a benchmark to difficulties in reproducibility, since there have been benchmarking efforts for statistical models. These issues can, in most cases, be attributed to the lack of shared data, code and trained model weights.

Due to the underexplored development of machine learning-based PRS models with respect to genomic sequence-related tasks, we have identified the need for a benchmarking and development framework to aid the development of models. We identify the following necessary properties for a framework based on the burden of data accessibility and reproducibility in machine learning model development:

**Flexibility**:

The framework needs to be able to operate on a multitude of different datasets, differing in size, population structure and including phenotypes of quantitative or categorical type. Additionally, the framework should be easy to use to promote its use in the research community, as well as easily extendable to new models and customisable for evaluation and data preparation.

**Comparability**:

To validate newly developed methods rigorously it is necessary to compare their prediction against state-of-the-art PRS models. Therefore, the framework should integrate benchmarking standards that allow for the immediate comparison across models, machine learning-based as well as established statistical models. Benchmarking standards have shown to be fruitful for the development of machine learning models in tasks such as image classification (Deng et al., 2009).

**Reproducibility**:

While the comparison against established PRS models yields insights about predictive performance, it is equally important to evaluate reproducibility of newly developed models. Reproducibility aids in validation of models across datasets as well as investigations about causality, and cross-ancestry transferability.

## 1.3 SCOPE OF THIS WORK

In this work we propose MLBF-PRS, the first benchmarking and development framework for machine learning-based PRSs. We make contributions to address the above mentioned requirements. Firstly, to address flexibility, our framework focuses on ease of usage and extendability through the use of Nextflow pipelines, which connect core functionalities and processing steps. This modular approach will promote the establishment of the pipeline as a valuable choice for model development and benchmarking. The framework offers a new way of benchmarking machine learning models on restricted datasets due to a stringent end-to-end model training and evaluation that allows for the immediate comparison of established PRS models to the machine learning model in development. Crucially, recently developed models often claim superior performance in their original publications (Jayasinghe et al., 2024). Therefore, to facilitate comparability across datasets, and to allow for quick validation of studies, all parameters associated with data preparation, model evaluation and model training are recorded. In this way, the results of an experiment can be reproduced by sharing configuration files if access to a dataset is granted. Apart from sharing of training configurations, the framework also promotes the sharing of trained model parameters to validate models without the need for data sharing, similar to the already established sharing of summary statistics of GWAS studies (Sollis et al., 2023). An in depth overview of novel contributions and its explicit usage inside the framework is discussed further in section 2.

## 2 THE DEVELOPMENT AND BENCHMARKING FRAMEWORK

The framework aims to promote the development of machine learning-based PRS models as an alternative to an open benchmarking dataset. To provide similar benefits as a benchmarking dataset, we incorporate flexibility, comparability and reproducibility together with PRS specific recommendations to present a state-of-the-art framework that fills the gap for machine learning-based PRS model development.

### 2.1 FLEXIBILITY

The framework consists of multiple modules that, when combined together in a single end-to-end pipeline, allow for the automatic training and evaluation of machine learning models. Each module consists of multiple scripts that can be customised or expanded to fit an algorithm's properties or include new algorithms. As visualised in Fig. 1 the three core modules are related to data preparation, which is known as quality control in the context of PRS, model training, and model evaluation. The core module for model training is separated into three different streams of models due to their specific requirements. Here, established statistical PRS models or tools can be trained or calculated, while machine learning models can be trained with a separate training pipeline for deep learning models.

One strength of the framework lies in its modularity. To include a new algorithm it is only necessary to create connection points in the form of scripts to the quality control module for input and the evaluation model for its output. Apart from adding new algorithms, it is also possible to include different or additional data preparation or augmentation steps at the connection points of each quality control step. Lastly, as the prediction results also report raw scores and effect sizes, any desired metric may be inferred from a new evaluation script.

The framework is built on Nextflow (Di Tommaso et al., 2017), a software tool that allows for the orchestration and execution of scripts as pipelines. Its advantage lies in the ability to run independent scripts in parallel while sequential steps wait for output from dependent scripts. This effectively allows for the execution of the three streams in the core module in parallel and for the training of models in parallel. However, this capability is restricted by available resources. To orchestrate parallel computing, Nextflow allows for the allocation of resources to a specific pipeline through configuration files. The statistical PRS tools implemented in this framework are all callable via

command line interface, which makes them easily implementable into a Nextflow pipeline. Python scripts and R scripts are all created as command line tools to make them implementable and capable for rapid testing.

The framework utilizes the PLINK file format (Purcell et al., 2007) as well as the PLINK software for quality control. We specifically choose to work with the PLINK file format as rich data sources like the UKB or All of Us (Bick et al., 2024) release genetic data in the PLINK file format. PLINK itself also offers commands to convert other file formats like .vcf into PLINK files. We make the data accessible for use in Python-based machine learning frameworks scikit-learn (Pedregosa et al., 2011) and Pytorch (Paszke et al., 2019) through specialised data loading modules.

## 2.2 COMPARABILITY

To facilitate comparability like an open benchmark, the framework utilizes the development and evaluation pipeline in such a way that, independent of the dataset, a comprehensive benchmark can be created. A dataset specific benchmark acknowledges the unique challenges encountered when developing phenotype-specific models. While the environment surrounding the PRS model development can be transferred between datasets (i.e. quality control, data loading, or statistical analysis) the phenotype specific modelling can differ significantly, because phenotypes differ in heritability, gene interactions or biological pathways.

Consequently, the evaluation and development framework offers an environment that follows recommendations for PRS-specific model evaluation. The data preparation module includes an extensive quality control proposed by Choi et al. (2020). The quality control is applied to summary statistics as well as the individual level data. Summary statistics are typically not used for the development of machine learning-based methods, but are still necessary for the statistical models considered for comparison. Apart from filtering SNPs and individuals by genotyping rate, sample missingness, Hardy-Weinberg equilibrium, heterozygosity, minor allele frequency and imputation score through the use of PLINK, the quality control pipeline includes the following additional steps: removal of ambiguous SNPs, resolving or removal of non-resolvable mismatching SNPs, removal of duplicate SNPs, removal of individuals with mismatching reported and genetic sex and lastly the removal of closely related individuals. Besides the quality control steps the data preparation modules allows for the calculation of principal components, which are commonly used in many PRS models to account for confounding effects through population structures in datasets. Lastly filtering of individuals, SNPs and data augmentation of covariates is also done in the first module of the framework. In contrast to Pain et al. (2021), we do not restrict the choice of SNPs to HapMap3, we intentionally leave this choice open due to flexible choices for machine learning-based models, but report on SNPs at the end of the pipeline.

Between the data preparation and model training lies an integral part of comparable model development which is data splitting into a training dataset and a hold-out dataset for later model evaluation. Similar to Pain et al. (2021) we integrate k-fold cross-validation.

Due to the majority of state-of-the-art PRS model development focusing on statistical models it is essential to compare any machine learning-based model against these established methods. Therefore, we integrate a representative collection of such methods to benchmark against the machine learning-based models. We chose to integrate PRSice-2 (Choi & O'Reilly, 2019), to represent the gold standard approach of clumping and thresholding (CT), an extension to CT called stacked CT Privé et al. (2019), LDpred2 Privé et al. (2021), Lassosum Mak et al. (2017) and Lassosum2 Privé et al. (2022), PRS-CS Ge et al. (2019) and PRS-CSx Ruan et al. (2022), SBayesRC Zheng et al. (2024) and PRSet Choi et al. (2023) to capture a variety from the landscape of current state-of-the-art solutions. Some of the models require additional data such as LD reference panels, pathway information or ancestry specific data, which is made available in the pipeline.

In contrast to the statistical models, the machine learning-based models are in most cases trained in a supervised setting to predict a phenotype. In the case of a binary phenotype, such as cancer status, the class likelihood is used as a PRS. In that case genotype data is used together with covariates for training. The data loader combines and provides this data for every individual during model training or evaluation. The covariates need to be specified otherwise all of them are used.

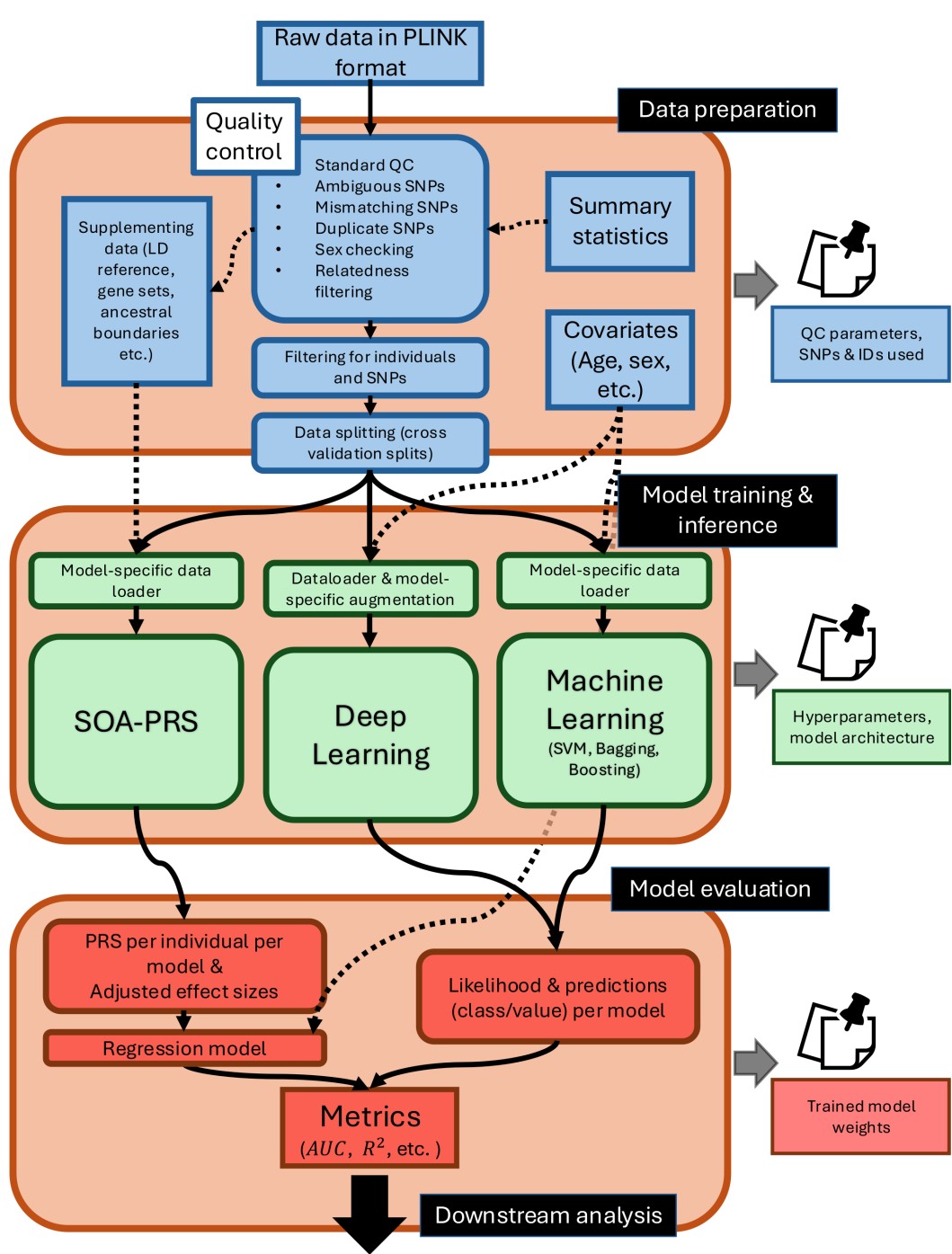

Figure 1: Visualisation of the framework, divided by the three core modules. Each module retains data to facilitate reproducibility.

To create a predictive model from the statistical models, typically, a linear, for continuous phenotypes, or a logistic, for binary phenotypes, regression model is fitted on the PRS together with covariates. Therefore, we report the PRS for each of those models per individual as well as the effect size for each SNP per model. With this raw data it is possible to then fit predictive models per method or even create ensemble models. From here metrics can be calculated to compare perfor-

mance across models. We recommend following reporting standards as proposed by (Wand et al., 2021). These include metrics such as $R^2$, AUC, reclassification indices but also statistical analyses like distributions of cases and controls.

In summary, the framework offers a standardised and concise end-to-end benchmarking approach for the development of new PRS models or inference of a selection of models which can be easily customised and extended.

### 2.3 REPRODUCIBILITY

An equally important problem in the development of PRS models lies in the validation of a model on external data which it has not seen during model training. To achieve such reproducibility the framework offers essentially two capabilities that aid the validation of models. Firstly, every parameter in the pipeline, from quality control, over model training, to evaluation, is specified in configuration files associated with each of the three modules, as shown in Fig 1. In this way, the whole pipeline or single steps should be reproducible if access to the same data resource is given. For instance, in favourable circumstances, where a dataset is openly accessible, e.g. available on dbGaP (Tryka et al., 2014), a validation study would only need to download the dataset and with shared configuration files an entire analysis could be easily repeated. A comprehensive overview of PRS development and calculation through these configuration files also promotes the clinical transferability of PRSs (Xiang et al., 2024). However, as previously mentioned, access to datasets might be restricted. In such cases we recommend opting for additional sharing of model architecture and trained weights in case of machine learning model development. The framework, by default, saves trained weights and each model definition is stored in separate files. Sharing a model in this way retains information in its trained weights and sharing of original data is not necessary, similar to sharing results of GWAS analyses in form of summary statistics or published PRSs on the PGS catalog (Lambert et al., 2021). Apart from the sharing of trained machine learning model weights, the modified effect sizes of each statistical model are saved as well. The sharing of learned machine learning model weights together with finetuning pretrained models remains an underexplored area in the field of PRSs and our framework makes it more accessible.

## 3 USAGE OF THE FRAMEWORK

As MLPBF-PRS is a modular framework it can be accessed on multiple levels from top to bottom. At the highest level a whole analysis across all core modules can be conducted by running the script `run_pipelines.sh`. This script then runs multiple Nextflow pipelines following the illustration of information flow shown in Fig. 1. Each pipeline can however be executed independently given its input data exists. As an example the model training and inference pipelines require input data generated from the quality control module. Pipelines are called through Nextflow as shown below in the case of the statistical PRS models or for deep learning-based PRS approaches:

```
nextflow run workflows/prs_models_pipeline.nf
            --params-file workflows/config/params_prs.yaml
nextflow run workflows/dl_prs.nf
            --params-file workflows/config/dl_config.yaml
```

Inside the `.yaml`-files model parameters are set to promote reproducibility. At the foundation of the framework lie Python scripts, R scripts and command line tools. The scripts or tools are embedded into Nextflow pipeline, but can also accessed for rapid testing. As an example the training script for deep learning models is called inside a Nextflow pipeline as shown below. The `params.xxx` inputs in this case are defined inside a `.yaml` file while another configuration file is passed to the training routine via `${params_file}`.

```
python ${params.base_dir}/bin/train_model.py \
    --genotype_file ${genotype_data} \
    --phenotype_file ${phenotypes} \
    --indices_file ${indices} \
    --params_file ${params_file} \
    --fold ${fold} \
```

```
                    --max_epochs ${params.max_epochs} \
                    --batch_size ${params.batch_size} \
                    --output_model model_fold_${fold}.pt \
                    --output_metrics metrics_fold_${fold}.json \
                    --output_log training_log_fold_${fold}.csv \
                    ${wandb_args}
```

The multi-levelled approach to the framework enables in depth as well as high level access to the field of PRS development.

## 4  DISCUSSION

### 4.1  RELATED RESEARCH

There exist several open benchmarks across genomic data problems that have successfully produced novel machine learning models. The precisionFDA Truth Challenge 2.0 Olson et al. (2022), which utilizes the Genome in a Bottle benchmark, motivated the development of new machine learning approaches to improve variant calling. CAGI (Jain et al., 2024) and CAFA (Zhou et al., 2019) aim to promote model development for prediction of genetic variant impact and protein function prediction respectively. In CAGI challenges machine learning models have shown to outperform statistical models Savojardo et al. (2019) and in CAFA3 a machine learning ensemble model performed best (Zhou et al., 2019). Recent efforts have evaluated genomic foundation models systematically. Patel et al. (2024) introduce a DNA language model (DNALM) benchmark for short, local sequences evaluating models like HyeanaDNA, DNABERT or Nucleotide Transformer. Similarly, Kao et al. (2024) evaluated DNALMs for long range dependencies. Liu et al. (2024) propose a benchmarking framework for DNA sequences focusing on modularity and reproducibility like MLBF-PRS. Notably, these models are all trained on openly available sequence data. Kopp et al. (2020) offer a flexible Python module, named Janggu, for sequence data management to train neural networks.

However, PRSs present unique challenges: integrating genome-wide signals rather than local sequences, predicting clinical outcomes rather than molecular phenotypes, and dealing with individual level data. While there is no shortage of development of machine learning-based PRSs there exist no comprehensive open benchmarks across such models. Benchmarks mostly focus on statistical models or treat machine learning models rather generically (Zhao et al., 2024; Wang et al., 2023; Allegrini et al., 2022), even though there exist a multitude of specifically designed neural network architectures (Schuran et al., 2025). These newly developed neural networks benchmark mainly against statistical models and not other task-specific neural networks, although the majority of studies report performance improvements of their respective novel method. For statistical models efforts have been made to make PRS models more accessible through pipelines that integrate multiple models. PGS-Depot focuses on a pipeline for summary statistic based PRSs (Cao et al., 2024). Yang & Zhou (2022), Lee et al. (2023) and Jin et al. (2025) provide a web based solution for polygenic risk prediction, addressing accessibility and ease of usage. Yaraş et al. (2025) and Pain et al. (2024) focus on a standardised and reproducible pipeline, where PGSXplorer also uses Nextflow pipelines. Pham et al. (2022) address diverse populations in their standardised benchmarking software. None of these existing pipelines investigates machine learning-based solutions.

Witteveen et al. (2022) propose a different approach in creating an open benchmark. Here, a synthetic dataset based on real genotypes is generated that yields privacy preserving open dataset. They show that absolute and relative performance of models can be accurately evaluated using statistical PRS models. Mendes et al. (2025) suggest that utilizing synthetic data can accelerate machine learning model development even for rare diseases.

### 4.2  MLBF-PRS TO BRIDGE THE GAP BETWEEN STATISTICAL AND MACHINE LEARNING-BASED PRS

We position MLBF-PRS in the existing gap between established benchmarking approaches from the machine learning community for DNALMs and the computational tools such as pipelines and frameworks from the bioinformatics domain. The focus of MLBF-PRS is utilizing the accessibility of a benchmarking pipeline to catalyse reproducible machine learning-model development. We

suspect current PRS benchmarks and frameworks ignore machine learning models due to a lack of reproducibility, therefore, our framework encourages the development of reproducible research. In this way provide a platform for machine learing-based models with the aim to benchmark them against another, while still providing statistical model predictions as a baseline.

### 4.3 LIMITATIONS AND FUTURE DEVELOPMENTS

Many analyses of PRS include the investigation of interpretability to find or validate known causal biological pathways. Explainable artificial intelligence has the potential to yield such insights from machine learning models (Novakovsky et al., 2023), therefore, we plan to integrate methods such as DeepLift (Shrikumar et al., 2017) or Grad-CAM (Selvaraju et al., 2020) into our framework. So far during development we tested our framework solely on synthetic data generated from a reference panel that used whole genome sequencing with high coverage (Byrska-Bishop et al., 2022). When dealing with real datasets, it is common to have to account for missing data, imputations, possible confounding factors and very imbalanced cases and controls. In the future we plan to apply the framework to large biobank data such as the UKB to test robustness of the pipeline and to further expand its capabilities. Perelygin et al. (2025) have shown superior improved of non-linear machine learning models on simulated epistasis phenotypes with respect to linear models, we plan to validate performance gains of non-linear models on real data in the future with MLBF-PRS. So far the framework lacks its own data generation pipeline, we think, that the privacy preserving data generation proposed by Witteveen et al. (2022) could further improve the framework. Consequently we plan to integrate automatic data generation functionality to further improve accessibility. Apart from that, we only integrate a collection of all the statistical models available, but the flexibility and modularity of the framework allows for the integration of additional models in the future. Lastly, we suggest that the reproducible pipeline could be used to enable privacy preserving model training through federated learning (Sabha, 2025).

## 5 CONCLUSION

Adoption of machine learning-based solutions for PRSs lags behind their statistical counterparts as well as the establishment of machine learning-based models as state-of-the-art models in other genetic data problems. Here, we identified three key factors that we believe hinder progress in machine learning-based PRS development, namely accessibility issues to valuable datasets, compromised comparability due to the diverse nature of datasets and the lack of an open benchmarking datasets, common in deep learning-model development, and finally a lack in reproducibility. Therefore, we introduce MLBF-PRS, a framework designed to make the development and inference of machine learning-based PRS models more accessible while providing an alternative solution to open benchmarking datasets. The framework acknowledges the diversity of existing data and aims to bridge the current gap in realizing the modeling advantages that machine learning approaches offer for SNP data analysis. MLBF-PRS is designed with extensibility in mind, making it easy for the research community to contribute enhancements and adopt the software to accelerate model development. Given that machine learning models have demonstrated promising results in cross-ancestry applications (Elgart et al., 2022) and variant discovery (Lakiotaki et al., 2023), we believe that machine learning solutions for PRS warrant further investigation and validation. MLBF-PRS serves as a practical tool to advance progress in this rapidly evolving field.

### LLM USAGE DECLARATION

Large Language Models have been used for editorial polishing (spelling and grammar checks) and as a coding assistant to generate and debug code.

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
