# OpenReview forum: "MLBF-PRS: A MACHINE LEARNING MODEL DE- VELOPMENT AND BENCHMARKING FRAMEWORK FOR POLYGENIC RISK SCORES"
_ICLR.cc/2026/Conference — ICLR 2026 Conference Withdrawn Submission_

### Official Review · Reviewer_WtM5 · 2025-10-26

**Soundness:** 1
**Presentation:** 2
**Contribution:** 1
**Rating:** 2
**Confidence:** 4

**Summary:**

The paper introduces MLBF-PRS, a modular Nextflow-based framework that standardizes end-to-end development and benchmarking of polygenic risk score (PRS) models. The objective is to address three major barriers: data accessibility, cross-study comparability, and reproducibility. It provides quality-control and data-preparation pipelines aligned with PRS best practices, parallel training/evaluation of ML models (e.g., SVMs, random forests, neural networks) alongside leading statistical PRS methods (e.g. pruning and thresholding, LDpred, etc), and rigorous experiment tracking that records configurations and trained weights so results can be replicated or validated on new cohorts without sharing raw data. The authors position MLBF-PRS as a practical alternative to open benchmarks for restricted datasets and report initial demonstrations on synthetic WGS-based data, outlining a roadmap to add explainable-AI methods (e.g., DeepLIFT, Grad-CAM), apply the pipeline to large biobanks like UK Biobank, and explore privacy-preserving data generation and federated learning.

**Strengths:**

Please refer to Weaknesses.

**Weaknesses:**

The topic of polygenic risk scores and a unified ML framework for developing them, is timely and clearly relevant to ICLR. However, the submission reads as an incomplete framework paper: it lacks empirical validation on real, widely available cohorts (e.g., UK Biobank), quantitative benchmarks across multiple phenotypes, ablation studies, and runtime/resource evaluations. Without end-to-end results and head-to-head comparisons against established PRS baselines (e.g., PRSice-2, LDpred2, etc), it is not possible to assess scientific merit or practical impact. I cannot recommend acceptance until rigorous experiments, transparent artifacts (configs, seeds, trained weights), and a clear demonstration of where ML approaches outperform statistical baselines are provided.

**Questions:**

Please refer to Weaknesses.

---

### Official Review · Reviewer_F9TJ · 2025-10-30

**Soundness:** 1
**Presentation:** 1
**Contribution:** 2
**Rating:** 2
**Confidence:** 4

**Summary:**

PRS evaluation pipeline with ability to train your own ML-based PRS scores.

**Strengths:**

1. Generally, it will be very convenient to have a pipeline where I can change one .yaml file, download one .vcf or .plink2 set and train dozens of PRS. The idea is sound and can be of great use to a community.

**Weaknesses:**

1. Related research section is all over the map, talking about benchmarks, competitions, etc. I suggest focusing on open-source evaluation and training frameworks for PRS and phenotype prediction. For example, you describe PGSXplorer and a few other frameworks/pipelines for PRS, but you are not comparing MLBF-PRS with them directly. After description of each of the similar frameworks, you should point out what they lack, and how MLBF-PRS addresses this.
2. Tested only on synthetic datasets, no reproducibility examples, i.e. take a well- known PRS, use the same dataset and QC params and show that your results match theirs closely.
3. No figures and tables comparing already implemented methods on at least synthetic dataset.
4. Train at least one SVM or DL PRS score and show results.
5. No nextflow or params.yaml examples in appendix.
6. No automated downloading of some sample datasets.
7. The paper fails to mention and compare its framework to pgsc_calc [1], the official Nextflow pipeline from the PGS Catalog. It is unclear why a new framework is needed rather than extending this tool that has been popular in the community.

[1] https://github.com/PGScatalog/pgsc_calc

**Questions:**

**Questions**
1. How easily can I customize QC part?
2. Can I run it on SLURM cluster or any other cluster?
3. Can it work with summary statistics AND individual-level genotype-phenotype data, i.e. UKB? Maybe I missed it.
4. What will happen if I want to use a very big dataset, or WGS data?
5. Is it a wrapper with some PRS scores implemented separately around PGS-calc tool or not?

**Feedback**
1. Too much text, too few other things. Some parts of the readme can go into appendix or main text to better illustrate what is going on inside MLBF-PRS and what it outputs.
2. Maybe this paper is better suited for ISMB open-source track, or some Datasets & benchmarks track.

---

### Official Review · Reviewer_dB77 · 2025-10-31

**Soundness:** 3
**Presentation:** 2
**Contribution:** 3
**Rating:** 6
**Confidence:** 4

**Summary:**

The manuscript introduces a standardized pipeline with code for polygenic risk score prediction, implemented in Nextflow. This framework facilitates reproducible and accessible analysis for machine learning researchers working on PRS score prediction, addressing its inherent complexity

**Strengths:**

1.	The problem of polygenic risk score (PRS) prediction is inherently affected by numerous potential confounders. The authors address this challenge by providing a state-of-the-art pipeline that streamlines data preparation out of the box. It is understatement to say that without the availability of these tools this topic is inaccessible to the vast majority of ML scientists
2.	The framework focuses reproducibility by storing all relevant variables in well-structured configuration files.
3.	A comprehensive selection of classical statistical methods is included by default

**Weaknesses:**

1.	The manuscript does not include, in its main body, a direct head-to-head comparison between different machine learning methods, nor between machine learning and traditional statistical approaches. Incorporating such an analysis would not only enhance the value of the work but also enable researchers to validate the pipeline’s performance by reproducing the reported results.
2.	The manuscript does not discuss the potential role of DNA language models in the context of PRS prediction, nor whether such models could be integrated within the proposed framework.

**Questions:**

1. Would it be possible for the authors to include an initial benchmarking section, at least on synthetic data, comparing the performance of the various methods supported by the package?
2. Can the authors discuss the potential application of machine learning methods like DNA language models that leverage variant embeddings or representations rising from tilling of the genome using windows?

---

### Official Review · Reviewer_8Yp5 · 2025-11-11

**Soundness:** 2
**Presentation:** 2
**Contribution:** 1
**Rating:** 0
**Confidence:** 2

**Summary:**

The paper introduces MLBF-PRS, a modular Nextflow framework that introduces a benchmarking standard for polygenic risk score (PRS) models. The contribution is infrastructural rather than methodological. It is a pipeline that standardizes data preprocessing and QC steps, data splits, model training across statistical and ML baselines, while logging all configurations / hyperparameters, model weights, and per-SNP effects for reproducibility. The authors argue that challenges around restricted data access (due to genomic privacy concerns) and inconsistent evaluation & baseline selection practices have been the main obstacle against the adoption of deep learning models in PRS estimation applications. They argue this pipeline is a possible alternative way to overcome such challenges.

While the effort to standardize benchmarking is commendable and the contribution could be of interest to bioinformatic communities, there are two main reasons why I do not recommend the acceptance of the contribution: (1) It is unclear what contribution, beyond the inclusion of standard ML models like regression, SVM etc, the proposal introduces compared to the related work listed, as this would be quite an incremental contribution and (2) I am highly skeptical of the assessment that the listed reasons are the main obstacles for deep learning model development in the area and this contribution would overcome such challenges.

**Strengths:**

**1.** The proposed pipeline standardizes multiple data QC and preprocessing steps, such as individual relatedness filtering, duplicate SNP filtering, non-genetic confounder correction, standardization of train/validation/test splits. This would help with cross-study comparability.
**2.** The pipeline automatically runs classical ML models (linear/logistic regression, SVMs, random forest etc), ensuring the comparison against appropriate baselines every PRS estimation study should include, if the proposed pipeline/framework is widely adopted.
**3.** Logging model parameters and configurations is a good step towards reproducibility.
**4.** The authors provide the source code for the pipeline in supplementary materials. Their pipeline presents a modular framework.
**5.** Discussion of including data generation pipelines in the future work (e.g. simulation scenarios based on public data like 1000 Genomes Project etc) sounds like a good plan for such a pipeline.

**Weaknesses:**

**1.** Firstly, it is unclear exactly how the proposed pipeline differs from the ones reported in the Related Works section (the PRS-related ones) other than the inclusion of classical ML baselines. Are there any other shortcomings of the existing PRS model evaluation pipelines/frameworks the model improves upon? If so, this is not clear from the discussion of the related works.

**2.** I am not convinced that data access and evaluation standardization are the main bottlenecks for deep learning applications in PRS estimation. A more fundamental obstacle is the difficulty of controlling ancestry-related confounding in nonlinear models under strong linkage disequilibrium (LD, i.e. the high covariance structure between genetic variants/SNPs). In high-dimensional genotype data, many features can be ancestry proxies; nonlinear learners (DNNs, boosted trees) can easily exploit these correlations, especially when LD induces complex covariance between ancestry-informative markers and truly causal loci. Standard fixes (PC covariates, global residualization) are often insufficient once interactions/epistasis or phenotype–covariate couplings enter the model, and they can even introduce bias under misspecification. This problem gets even more non-trivial when one considers cross-ancestry heterogeneity in allele frequencies, population-specific nature of rare variants etc. The manuscript claims to advance the analysis of causal variants and cross-ancestry transferability through this framework but it is entirely unclear how it tackles this very difficult open challenge.

**3.** It is unclear to me how the proposed framework is a replacement for data sharing. Even if the model weights and configurations are shared across teams through this framework, if the two teams don't have access to the same data, it means they will be testing the model on genetic data with different population structure. Especially with the lack of details on how population stratification etc is done in QC (or if it is even considered at all), I think this obstacle remains without much ease from the adoption of this framework.

Additionally, I have some minor comments on the writing:
The Introduction section discusses linear regression etc models as if they are not "machine learning" models but these do count as ML models. I believe you are actually referring to deep learning models when you say "state-of-the-art models in PRS estimation do not rely on machine learning algorithms". Additionally, the following statement requires some clarification: "These aforementioned genomic data machine learning models use genome sequence data, which differs quite significantly from the genotype array data used for PRS development, for which the data consists of single measurements of variations at single loci across the whole genome". One could use genome sequence data (e.g. whole genome sequencing data) for PRS, as well. I believe the distinction you intended to make is between public reference genomes and/or functional genomics data without privacy concerns vs the need for individual-level genomic data (which comes with privacy concerns) in the PRS applications. The sentence reads as if the difference is due to genome sequencing vs SNP array technologies while that is not the case.

**Questions:**

**1.** In the Related Works section, you say "Yaras¸ et al. (2025) and Pain et al. (2024) focus on a standardised and reproducible pipeline, where PGSXplorer also uses Nextflow pipelines", which sounds as if Nextflow-based pipelines to standardize PRS estimation method benchmarking already exist. What are the shortcomings of these models that your pipeline addresses? The only statement regarding this is "None of these existing pipelines investigates machine learning-based solutions", but your own pipeline requires users to integrate their own machine learning model via scripts to connect to the Nextflow pipeline. Do you mean your pipeline is the only one that automatically includes established ML baselines such as linear/logistic regression, SVMs, random forests etc? If so, this is an incredibly incremental contribution. If not, the novelties should be more clearly detailed.
**2.** How do you handle population stratification and ancestry-based confounder correction in data preprocessing, especially in the cases of benchmarking non-linear models? You mention enabling cross-ancestry transfer on page 3, but without such details, it is unclear how this would be achieved, which is a big non-trivial open problem in the PRS estimation field.
**3.** In Figure 1, what is the difference between the dashed lines/arrows and the solid lines/arrows?

---

### Note · Authors · 2025-11-27

**Comment:**

We would like to thank the reviewers for the valuable recommendations for the manuscript. Following the comments we have deciced to withdraw the submission and submit a reworked version to a different conference at a later stage.

**Withdrawal Confirmation:**

I have read and agree with the venue's withdrawal policy on behalf of myself and my co-authors.